# Psychometric Properties of the Work Ability Index in Health Centre Workers in Spain

**DOI:** 10.3390/ijerph182412988

**Published:** 2021-12-09

**Authors:** Inmaculada Mateo Rodríguez, Emily Caitlin Lily Knox, Coral Oliver Hernández, Antonio Daponte Codina

**Affiliations:** 1Department of Social and Organizational Psychology, Faculty of Psychology, Universidad a Distancia de Madrid (UNED), 28040 Madrid, Spain; 2Escuela Andaluza de Salud Pública, 18011 Granada, Spain; emily.knox.easp@juntadeandalucia.es (E.C.L.K.); antonio.daponte.easp@juntadeandalucia.es (A.D.C.); estar-group.easp@juntadeandalucia.es; 3CIBER de Epidemiología y Salud Pública (CIBERESP), 28029 Madrid, Spain; 4Department of Social, Work and Differential Psychology, Faculty of Psychology, Universidad Complutense de Madrid, 28040 Madrid, Spain; maoliver@ucm.es

**Keywords:** work ability index, Spain, validation, healthy ageing, cohort study

## Abstract

The aim of the present study is to analyse the psychometric properties of the work ability index (WAI) within a sample of Spanish health centre workers. The WAI was translated into Spanish using transcultural and forward–backward translation processes and administered to 1184 Spanish health centre workers. Internal consistency, predictive validity, and discriminative ability were examined. Exploratory factor analysis, via principal components analysis and confirmatory factor analysis, determined the most appropriate questionnaire structure. All indices in relation to predictive validity and reliability were acceptable. Exploratory factor analysis supported validity of the one-factor structure, however, confirmatory factor analysis suggested better properties in relation to a two-factor structure (χ^2^ = 59.52; CFI = 0.98; TLI = 0.96; RMSEA = 0.06). Items 3, 4, and 5 loaded onto factor one, and items 1, 2, 6, and 7 loaded onto factor two. The two factors could be broadly described as “subjectively estimated work ability” and “ill-health-related ability”. The WAI is valid and reliable when administered to health centre workers in Spain. In contrast to that suggested by studies conducted in other countries, future research and practical application with similar respondents and settings should proceed using the two-factor structure.

## 1. Introduction

The workforce is ageing in many countries around the world [1,2]. The challenges this presents have given rise to government policies with a stronger emphasis on work ability promotion and the extension of working life [3].

Work ability is defined as the worker’s ability to perform their work based on their mental and physical conditions and occupational needs, in addition to the capacities and capabilities of individuals in relation to the physical and psychological needs of work [4]. In the present study, we defined work ability as the product of self-evaluations of the balance that exists between the psychophysical demands of the job at hand and the individual’s psychophysical capacity [5].

Promoting and maintaining good work ability in all phases of working life is vital as poor work ability has been linked with an increased risk of poor work quality, sickness absence, long-term disability, and early retirement [6,7,8]. On the other hand, excellent work ability is closely related with improved work quality, and improved quality of life and well-being amongst employees, thereby decreasing the likelihood of early retirement and absenteeism, whilst increasing motivation and productivity in employees of all ages [9,10].

For all of the above, work ability monitoring by employers could help preserve the health and well-being of employees, whilst also informing interventions targeting the positive health and motivation of workers [11].

The most commonly used method for assessing self-rated work ability is the work ability index (WAI), which was developed by researchers of the Finish Institute of Occupational Health [12].

The WAI consists of a combination of self-assessments of work ability, diagnoses, symptoms, and sickness absence. It is intended to measure a single dimension of work ability, that is, the health and functional capacity dimension of work ability [4]. Predictive validity of the WAI has been tested in several prospective studies, commonly with long-term sickness absence included as an outcome [13].

Having an instrument like the WAI to measure the work ability of workers could help with early detection of potential physical and mental health impairments. Implications of this on group or individual work ability may help organisations to develop programs and services for prolonging work ability [14].

Studies have shown that the WAI is a helpful tool for predicting long-term sickness absence [6] and early retirement [15] and identifying prognostic factors for mortality and work disability [12,16].

This instrument has now been translated into a number of languages and validated in a number of countries worldwide (e.g., [13,17,18,19,20,21]). The WAI has been shown to constitute an appropriate option for evaluating work ability in a general population sample (*n* = 1786) in Sweden [13], 3049 heavy industry employees in Greece [17], 3084 health workers in Brazil [18], 3968 nationally representative employees in Germany [19], 750 Persian car manufacturing and petrochemical workers [20] and in manual and office workers in Holland [21], although in the latter, cut-points differed in the two types of examined workers. The instrument has also been validated in Spanish speaking countries, including in a sample of 360 hospital and university workers in Chile [22], 100 primary health care workers in Argentina [23], and 24 hospital workers in Cuba [24]. In Spain, it has been validated by the INNST in a sample of 302 workers from different professional groups [25]. These psychometric studies have obtained different factorial structures for the instrument which pertain to 1, 2, or even 3 factors. In fact, one of these studies [21] found two different structures could be used within their sample depending on their occupational category. Thus, it is possible that factors such as the local context in which it is applied, the occupational sector, and other factors such as age and sex are responsible for these differences in psychometric outcomes. The disperse outcomes of the previously conducted studies highlight the need for the WAI to be validated, not only generally in relation to the country of administration but, also, within the specific occupational group of interest.

The purpose of the present study was to assess the psychometric properties of the Spanish version of the WAI in a Spanish sample of healthcare workers, examining the scale’s internal consistency, predictive validity, and factorial structure.

The overall aim of this is to provide the Andalusian health service with a valid version of the WAI, which can be used to measure the work ability of workers in the health sector at a national level and enable the detection of potential health impairments at early stages. The present study constitutes the first phase of a cohort study being conducted within the Andalusian health service. The resulting instrument, together with other measures, will be used in later studies to track health outcomes in this group.

## 2. Materials and Methods

Cross-cultural adaptation procedure: The 2nd revised edition of the WAI [26], which consists of 7 items, was used in this study. The WAI questionnaire was translated from its original English version by two bilingual translators through a process of back-translation [27]. A preliminary version was piloted within a sample of 30 individuals with the aim of examining comprehension and acceptance of each question. Comments made during the pilot were evaluated by an expert panel in order to develop the final questionnaire (full process detailed in Appendix A). The final questionnaire underwent psychometric analysis, with results presented in the present study.

Psychometric validation procedure: All procedures were approved by the research ethics committee.

Participants: Workers from six public health centres attending the occupational health department of participating health centres (three hospitals, two primary health care centres, and a combined health facility) to complete a routine health check were contacted. A member of the research team working at each centre explained the aims, procedures, and ethical considerations of the study. Participants were included in the study on a voluntary basis after signing an informed consent form. In total, 1184 workers (70% of workers invited to participate) answered the questionnaire. Data collection was conducted throughout 2019. Workers were excluded from the study if they had any other reason for attending the centre other than a routine health check-up.

Instruments: The version of the WAI obtained following a process of transcultural adaptation to the Spanish context was used. This comprises seven indicators. Indicators WAI1, WAI3, WAI4, WAI5, and WAI6 pertain to one item each, WAI2 comprises two items, and item WAI7 pertains to three items. As seen in Appendix A, different response formats and scales are used for each indicator. Thus, outcomes for the seven indicators are transformed before being summed to produce an overall score. Final scores range between 7 and 49, with higher values indicating better work ability. The questionnaire can be consulted in Appendix A.

In addition to completing the WAI questionnaire, participants were also administered other self-completion measures such as the SF-12 [28]. The SF-12 measures self-perceived health via 12 questions which are equally divided between a physical component and a mental component. A standardised z-score for each component is produced which ranges between 1 and 50. This procedure has been described and validated with Spanish samples [29].

Analysis: The tool was scored continuously as opposed to categorically, as proposed by Bethge and colleagues [30]. Internal consistency of the Spanish version of the WAI was examined through Cronbach’s *α*. Kendall’s tau b correlations were also conducted as data did not fulfil assumptions of normality. Correlations were performed between scale items to examine whether items were consistent in pertaining to the same general construct. Correlations were also performed between the overall WAI score, and overall and sub-component SF-12 scores. SF-12 was chosen as the standard criterion because of its ability to evaluate overall health based on the subject’s own perception. It is also a widely used instrument in health surveys and has been validated in a number of countries worldwide, including in Spain [31].

Concurrent validity of the WAI was examined by constructing a linear regression model to examine whether WAI scores reasonably predicted SF-12 scores. Given that health is a strong predictor of work ability, we expect WAI scores to significantly predict SF-12 scores.

Discriminative ability was determined in two ways. Firstly, area under the ROC was examined. The ROC was produced from a regression model including WAI and SF-12. This is used to establish whether WAI scores correctly identify health groups. Secondly, health groups were divided into ‘good’ and ‘poor’ health groups according to a median split of SF-12 scores. T-test analysis was then conducted, comparing WAI scores between health groups.

Construct validity was examined using exploratory factor analysis, through principal components analysis with direct oblimin rotation, followed by the maximum likelihood method. The whole sample was used in both analyses. The resultant structure was then verified using confirmatory factor analysis. All potential two-factor combinations were tested, as was the model indicated by exploratory factor analysis. Analysis was conducted for the overall sample and repeated for age, gender, and educational sub-groups to examine variance across important demographic groups. The statistical software package SPSS AMOS version 26.0 was used to conduct confirmatory factor analysis, whilst SPSS version 26.0 was used for all other analysis.

## 3. Results

This section may be divided by subheadings. It should provide a concise and precise description of the experimental results, their interpretation, as well as the experimental conclusions that can be drawn.

### 3.1. Demographic Data

A total of 1184 working adults were recruited to participate, of which 1062 completed all WAI items and 1056 completed all SF-12 items. The majority of the sample was female (71.1%), had a university degree (59.4%), lived with a partner (with or without other relatives; 73.9%) and worked in nursing, physiotherapy, or another associated profession (29.5%). On average, participants were aged 56.5 years (SD = 5.7) and worked 34 h a week (SD = 11.7).

Average WAI scores were as follows: item 1 = 8.04 (SD = 1.71); item 2 = 7.98 (SD = 1.53); item 3 = 4.04 (SD = 2.16); item 4 = 5.13 (SD = 1.12); item 5 = 4.24 (SD = 1.22); item 6 = 6.06 (SD = 1.73); item 7 = 3.18 (SD = 0.80); overall = 37.77 (SD = 7.20). Participants reported an average standardised health score of 59.41 (SD = 16.97), with scores of 30.01 (SD = 10.29) and 29.35 (SD = 8.54) for the physical and mental components, respectively.

### 3.2. Internal Consistency

Analysis of the Spanish version of the WAI produced a Cronbach’s α of 0.81, demonstrating acceptable internal consistency of the instrument. Elimination of any item from the scale reduced the Cronbach’s α, suggesting that all items positively contribute to its internal consistency. Internal consistency amongst items was also supported through satisfactory Kendall’s tau b coefficients between items (Table 1). Correlations between the WAI and SF-12 produced Kendall’s tau b values of 0.57 overall, 0.57 for the physical scale and 0.46 for the mental scale, with all having a *p* < 0.001. This also indicates strong internal consistency.

### 3.3. Predictive Validity

A linear regression model was constructed to predict SF-12 scores from WAI scores. Results indicate that the WAI shows strong predictive validity (R2 = 63.2; F = 1204.85; *p* < 0.001). When health was divided into its physical and mental components, WAI was found to significantly predict both aspects (mental: β = 0.32; *p* < 0.001, physical: β = 0.57; *p* < 0.001).

### 3.4. Discriminant Validity

Examination of the area under the ROC curve indicated the probability that the WAI correctly identifies health groups. Respondents were divided into two groups according to the median self-reported health score (‘good’, *n* = 516; ‘poor’, *n* = 540). The area under the ROC value was 0.87 (95% CI = 0.85–0.90), showing good discrimination. T-test analysis also supported the discriminative ability of the instrument, with respondents self-reporting good health ($\tilde{x}$ = 42.92 ± 3.28) also reporting significantly higher WAI scores than those reporting poor health ($\tilde{x}$ = 34.61 ± 6.95; t = 21.19; *p* = 0.000).

### 3.5. Construct Validity

Bartlett sphericity analysis produced a *p*-value of 0.000. This shows significant diversion between the correlation and identity matrix of the variables in the dataset. Thus, principal components analysis was an appropriate approach (Table 2). Kaiser–Meyer–Olkin analysis produced an acceptable value of 0.85. Principal components analysis suggested the presence of one single factor, with all items explaining at least 55% of variance. The least variance was explained by item 5, with this explaining 55% of variance, whilst the most explanatory item was item 4 (82%). This one-factor model explained 50.49% of overall accumulated variance. Confirmatory factor analysis examined appropriateness of this one-factor structure. Fit of the model was examined according to chi-square, however, given that this value is expected to be significant (indicating poor fit) when applied to samples > 200, other indices were also used. Comparative fit index (CFI), normalised fit index (NFI), Tucker–Lewis index (TLI), and RMSEA were also examined. For CFI, NFI, and TLI, values >0.90 indicate acceptable fit. In the case of RMSEA, values of 0.05–0.08 are acceptable and values <0.05 are excellent. As expected, the chi-square index produced was unacceptable (*p* = 0.000). The one factor model showed acceptable fit in terms of CFI (0.92) and NFI (0.92), however, TLI (0.836) and RMSEA (0.111) values were not acceptable. All items showed acceptable CFA loadings (>0.45) within the model. Factor loadings were as follows: item 1 = 0.78; item 2 = 0.80; item 3 = 0.51; item 4 = 0.79; item 5 = 0.50; item 6 = 0.54; item 7 = 0.60.

Alternative two factor models were examined to provide further assurance that the one-factor structure is the most appropriate. Table 3 presents goodness of fit statistics for the one-factor structure and the six two-factor models showing best fit. It can be seen that only one of these models produced acceptable fit parameters for all indices, outperforming the one-factor model (model D: χ^2^ = 59.52; CFI = 0.98; TLI = 0.96; RMSEA = 0.06). Model D also produced acceptable parameters for sub-populations defined according to gender, age, and education, with the only unsatisfactory index being an inflated RMSEA in relation to individuals with lower education. In this second model, factor 1 incorporated items 3, 4, and 5, and factor 2 incorporated items 1, 2, 6, and 7. Again, all items showed acceptable loading (factor 1: item 3 = 0.57; item 4 = 0.91; and item 5 = 0.52; factor 2: item 1 = 0.82; item 2 = 0.84; item 6 = 0.54; and item 7 = 0.60).

## 4. Discussion

The present study sought to verify the psychometric properties of the work ability index [16], translated into Spanish and administered amongst a large sample of Spanish health centre workers. Work ability is a hugely important concept in the field of social sciences as it has far-reaching and wide-ranging health, social, and economic implications [32,33,34]. Its evaluation is especially important within the framework of intervention development for promoting healthy aging at work [9].

Both the physical and mental dimensions of health, as measured by the SF-12 questionnaire and general health overall, were significantly and positively related with WAI scores. This provides evidence of predictive validity and discriminative ability within a sample of Spanish health centre workers. This is consistent with the theoretical framework of work ability, which is represented as health based on functional ability and the presence of disease.

Internal consistency was seen to be supported to a similar extent to that seen in previously conducted studies. The Cronbach’s α was slightly higher than that reported in a sample of Iranian nurses and healthcare workers [35], Iranian car manufacture workers [20], Cuban health workers [24], and workers from any sector in Spain [25]. In contrast, it was slightly lower than that seen amongst Jordanian nurses [36]. Further, a study conducted with a Greek sample demonstrated similar inter-item reliability [17].

Discriminative ability was slightly higher than that seen in Dutch workers and in a general population sample from Sweden, when using long-term sickness absence for comparison [13,21]. Tavakoli-Fard, Mortazavi, and Kuhpayehzadeh [37] also found the WAI to discriminate between health groups amongst Iranian female workers.

Predictive validity of the WAI has also been examined in other countries. In the same way that WAI significantly predicted self-reported health (SF-12) in the present study, WAI was associated with physical health and, to a lesser extent, mental health in a previous Spanish study [25]. Likewise, WAI has been seen to predict morbidity and comorbidity frequency, and sick-leave duration in Greek workers [17], and job strain in Iranian petrochemical workers [38]. A study by Rostamabadi and colleagues [39] amongst Iranian farm workers also found the WAI to predict self-rated health (assessed via the SF-36).

Many previous studies with German, Greek, and Brazilian samples [17,18,19,40] have dismissed a one-factor model of the WAI as inadequate. The present study also raises questions about the adequacy of the one-factor structure of the WAI within a large sample of Spanish health centre workers, however, findings are not entirely conclusive. The one factor model explained 50.5% of the overall variance, this being greater than that explained by the one factor model developed in a prior Spanish study (which explained 39.1% of variance) [25]. Other studies have previously reported greater explained variance for three-factor structures (53.9% [18], 56.6% [22], and 66% [23]). Indeed, all potential two-factor models were tested and whilst the majority also failed to demonstrate acceptable psychometric properties, one appears to show better fit to the empirical data than the one-factor structure. The two-factor model examined in the present study consisted of a factor formed by items 3, 4, and 5, and a second factor formed by items 1, 2, 6, and 7. This model produced superior fit statistics and proved acceptable for use within gender-, age-, and education-specific populations. The two factors could be broadly described as “subjectively estimated work ability” and “ill-health-related ability”, respectively. Perusal of factor loadings suggests that both the one-factor and two-factor models are appropriate, with all items loading acceptably within both models. Freyer et al. [19] reported the same structure within a sample of German employees. Another study also found the same structure within a sample of Brazilian nurses [41], with the only exception being that item 6 loaded with items 3, 4, and 5. Similarly, Abdolalizadeh et al. [35] examined an Arabic version of the WAI amongst an Iranian sample and reported a factor structure consisting of items 3, 4, and 5, and a factor structure of 1, 2, and 6. In contrast to the present study, item 7 loaded onto an additional third factor. Differences may be partly explained by the fact that in their study, item 7 was broken down into its three subcomponents, whereas in the present study it was entered as one item. Alexopoulos et al. [17] examined a Greek version of the WAI and reported the same two-factor structure supported in the present study. However, items 6 and 7 were seen to load onto both factors, which was not the case in the present study. In the present study, items 6 and 7 clearly loaded onto factor 2 only. Within a Brazilian sample, Cordeiro et al. [18] also found items 3, 4, and 5 (and 6) to group together, though their model had a three-factor structure. Finally, in a study conducted with 302 Spanish workers from any sector, a two-factor model also fitted best to the empirical data, although item 6 loaded on factor 1 [25].

The present examination suggests that analysis of the WAI, when administered to health sector workers in Spain, should be based on a two-factor structure. Although the one-factor structure produced acceptable loadings, it demonstrates questionable fit, suggesting that it may be obsolete.

Different studies have shown that low levels of work ability, measured through the WAI, are associated with long-term sickness absence [42], intentions to retire early [43], early exit from work [15], and other health outcomes [44].

For these reasons, this instrument enables organisations to estimate the current work capacity and future potential of their workforce, whilst also identifying risk groups who present low WAIs at early stages. Further, it provides a measure of the effect of the interventions developed to increase the work ability of workers.

The objective of this study was to examine the psychometric properties of the WAI when administered in health centres and the sample was, therefore, restricted to that setting, limiting generalisability of the findings. Future studies should also consider the effect of different working demands and skill requirements within healthcare workers. Additionally, the sample consisted mainly of women, however, this sex distribution is similar to that observed in health workers in the Spanish population [45]. Examination of non-responders also reveals a potential limitation, non-responders were slightly older, on average, and tended to be less educated than those who did respond. Investigation is warranted into the reasons for this, and future studies should strive to capture these individuals. This being said, non-response was low at below 10%. Finally, the study was cross-sectional and so conclusions regarding the tool’s test–retest reliability cannot be made.

Recruitment was done through occupational health appointments. In Spain, periodic health checks on all workers are mandatory in all health centres. However, one limitation of the present work is that some workers may not have been captured if they chose not to attend or could not attend their appointments. Nonetheless, the selection of study participants through occupational health appointments is a strength of this study, since such activities constitute the main context in which the WAI should be applied and produce the results that will lay the foundations from which preventive measures will later be developed.

## 5. Conclusions

The present study provides evidence that the Spanish version of the WAI is psychometrically reliable and valid for use within Spanish health centre workers. In similar samples and settings, instrument outcomes should be reported according to a two-factor structure (“subjectively estimated work ability” and “ill-health-related ability”). Additionally, WAI significantly predicted self-reported health (SF-12) in the present study.

Additionally, the WAI, as examined in the present study, demonstrated adequate psychometric properties for the different gender, age, and education groups assessed. This is useful as it may be relevant in certain settings to examine whether differences in work ability are driven by such characteristics.

Finally, our results suggest the usefulness for the WAI to be validated, not only generally in relation to the country of administration but, also, within the specific occupational group of interest.

## Figures and Tables

**Table 1 ijerph-18-12988-t001:** Inter-item correlations using Kendall’s tau b coefficient.

Item	2	3	4	5	6	7
1	0.19	0.26	0.39	0.24	0.33	0.33
	0.00	0.00	0.00	0.00	0.00	0.00
2		0.12	0.13	0.06	0.12	0.18
		0.00	0.00	0.06	0.00	0.00
3			0.49	0.29	0.27	0.20
			0.00	0.00	0.00	0.00
4				0.34	0.40	0.38
				0.00	0.00	0.00
5					0.18	0.19
					0.00	0.00
6						0.33
						0.00

WAI items: (1) Current work ability compared to lifetime best. (2) Current work ability in relation to work demands. (3) Number of current diseases diagnosed by physician. (4) Estimated work impairment due to disease. (5) Sickness absence during the past year (12 months). (6) Personal prognosis regarding work ability two years from now. (7) Psychological resources.

**Table 2 ijerph-18-12988-t002:** Exploratory factor analysis of the Spanish version of the work ability index administered to health sector workers.

Items	Principal Components Analysis *	Principal Components Analysis *	
	Factor	Factor	
	1	1	2
1. Current work ability compared with lifetime best	0.79	0.79	
2. Current work ability in relation to its demands	0.81	0.81	
3. Number of current diseases diagnosed by a physician	0.57	0.57	0.60
4. Estimated work impairment due to disease	0.82	0.82	
5. Sick leave during the past year (12 months)	0.55	0.55	0.55
6. Own prognosis of work ability two years from now	0.67	0.67	
7. Mental resources (feelings of joy, alertness, or optimism)	0.71	0.71	
Variance of the component (%)	50.49	50.49	13.41

* Rotation method: Oblimin with Kaiser normalisation.

**Table 3 ijerph-18-12988-t003:** Confirmatory factor analysis results—fit of various models.

	All Employees *n* = 1086	≤55 Years Old *n* = 593	≥56 Years Old *n* = 519	Female *n* = 796	Male *n* = 324	University Degree *n* = 470	No University Degree *n* = 350
One factor							
Chi-square	219.61	99.48	111.52	156.51	64.03	113.00	117.22
*p*	0.000	0.000	0.000	0.000	0.000	0.000	0.000
CFI	0.92	0.91	0.93	0.92	0.92	0.93	0.90
Tucker–Lewis	0.84	0.82	0.85	0.83	0.84	0.86	0.80
RMSEA	0.11	0.10	0.12	0.11	0.11	0.10	0.12
Two-factor							
A							
Chi-square	179.76	81.52	110.67	126.61	58.18	86.801	103.91
*p*	0.000	0.000	0.000	0.000	0.000	0.000	0.000
CFI	0.93	0.93	0.93	0.93	0.93	0.95	0.91
Tucker–Lewis	0.86	0.84	0.85	0.86	0.85	0.89	0.81
RMSEA	0.10	0.09	0.12	0.11	0.10	0.09	0.12
B							
Chi-square	219.43	99.46	110.43	154.71	63.64	112.10	116.27
*p*	0.000	0.000	0.000	0.000	0.000	0.000	0.000
CFI	0.92	0.91	0.93	0.92	0.92	0.93	0.90
Tucker–Lewis	0.82	0.80	0.84	0.82	0.83	0.85	0.78
RMSEA	0.12	0.11	0.12	0.12	0.11	0.11	0.13
C							
Chi-square	124.50	50.29	74.19	93.32	44.36	62.30	76.54
*p*	0.000	0.000	0.000	0.000	0.000	0.000	0.000
CFI	0.96	0.96	0.95	0.95	0.95	0.97	0.94
Tucker–Lewis	0.90	0.92	0.90	0.90	0.89	0.93	0.87
RMSEA	0.09	0.07	0.10	0.09	0.09	0.08	0.10
D							
Chi-square	59.52	28.48	35.18	52.37	21.98	23.89	51.73
*p*	0.000	0.008	0.001	0.000	0.056	0.032	0.000
CFI	0.98	0.98	0.98	0.98	0.97	0.99	0.96
Tucker–Lewis	0.96	0.97	0.96	0.95	0.97	0.98	0.92
RMSEA	0.06	0.05	0.06	0.06	0.05	0.04	0.08
E							
Chi-square	90.35	46.34	45.57	59.81	40.05	45.95	57.16
*p*	0.000	0.000	0.000	0.000	0.000	0.000	0.000
CFI	0.97	0.97	0.98	0.97	0.96	0.98	0.96
Tucker–Lewis	0.93	0.92	0.95	0.94	0.91	0.95	0.91
RMSEA	0.07	0.07	0.07	0.07	0.08	0.06	0.09

Model structures A–E: (A) 1, 3 versus 2, 4, 5, 6, 7; (B) 1, 6, 7 versus 2, 3, 4, 5; (C) 3, 4 versus 1, 2, 5, 6, 7; (D) 3, 4, 5 versus 1, 2, 6, 7; (E) 3, 4, 5, 6 versus 1, 2, 7.

## Data Availability

The data used for this work will be made available from the corresponding author upon reasonable request.

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
