# Peer review of "Psychometric Properties of the Work Ability Index in Health Centre Workers in Spain"

_ijerph, 2021, doi:10.3390/ijerph182412988_

Round 1

Reviewer 1 Report

The authors present a study to analyze the psychometric properties of the Work Ability Index (WAI) in a sample of Spanish healthcare workers. 
Although the work is interesting, it has some flaws. There is no literature review section. The authors should elaborate an exclusive section where they refer to previous WAI studies and their main conclusions. An interesting way to do this would be to add a summary table. The authors make a brief review of the literature in the conclusions section. They should do it beforehand, and then compare their results with those of the literature in the discussion section. The conclusions section is very poor and should be expanded. You could also add possible future lines of work and limitations of this work. Reference 45, Government of Spain (Gobierno de España) is written in Spanish, correct it.

The aim of the present study is to analyse the psychometric properties of the work ability 13 index (WAI) within a sample of Spanish health centre workers. The WAI was translated into Spanish 14 using transcultural and forward-backward translation processes and administered to 1,184 Spanish 15 health centre workers. 

Additional comments:

The paper is novel, has a clear and concise structure, and is suitable for an impact journal. Although the work is interesting, it has some flaws. There is no literature review section. The authors should elaborate an exclusive section where they refer to previous WAI studies and their main conclusions. An interesting way to do this would be to add a summary table. Authors should include current bibliography and relevant journals. The methodology is sound and correct, although the authors should explain it in more detail, so that any reader can understand it. Exploratory factor analysis, via principal components analysis and confirmatory factor 17 analysis, determined the most appropriate questionnaire structure.  The authors make a brief review of the literature in the conclusions section. They should do it beforehand, and then compare their results with those of the literature in the discussion section. The conclusions section is very poor and should be expanded. You could also add possible future lines of work and limitations of this work. The references are appropriate, but they should add some more current references to support the revision of the literature, which should have, as I said before, a specific section in the text after the introduction.

Author Response

We appreciate the comments and have made efforts to address the points made as best as possible.

Point 1: There is no literature review section. The authors should elaborate an exclusive section where they refer to previous WAI studies and their main conclusions. An interesting way to do this would be to add a summary table. The authors make a brief review of the literature in the conclusions section. They should do it beforehand, and then compare their results with those of the literature in the discussion section.

Response: We appreciate the reviewers comment and have made efforts to address the points made as best as possible. The third last paragraph (lines 64-82) of the introduction is dedicated entirely to summarising previous WAI studies and their main conclusions, with greater depth, specifically, information pertaining to the factor structure, being provided later in the discussion where it is more relevant. We have extended this section (see lines added line 65-73, lines 76-77 and lines 80-82) to provide greater details about the samples and most relevant information within word restrictions. A summary table would be more appropriate to a systematic review which is beyond the scope of the present study.  

Point 2: The conclusions section is very poor and should be expanded. You could also add possible future lines of work and limitations of this work.

Response: Additional future lines of work and limitations have been added on lines 316-317 and lines 319-323. Conclusions have been expanded on lines 337-341.

Point 3: Reference 45, Government of Spain (Gobierno de España) is written in Spanish, correct it.

Response: This reference has been corrected.

I attach the version of the article with all the suggestions

Thank you very much . Best regards

Reviewer 2 Report

It was a real pleasure to read such a well presented report.  The analysis and level of investigation was impressive.  I did feel that the over-representation of females may affect the ability for results to be generalised to all workers. It would be very valuable to see if your best two factor structure still applies when males are better represented. While more females may work in the healthcare industry that is not true of most occupations although I accept your focus was on healthcare workers in this instance.  You do need to address the non responders - who were they?  As they were recruited by the research team you should at least be able to group in gender and maybe healthcare type.  This would strengthen your conclusions.

Author Response

We appreciate the comments and suggestions

Point 1: I did feel that the over-representation of females may affect the ability for results to be generalised to all workers. It would be very valuable to see if your best two factor structure still applies when males are better represented. While more females may work in the healthcare industry that is not true of most occupations although I accept your focus was on healthcare workers in this instance.  You do need to address the non responders - who were they?  As they were recruited by the research team you should at least be able to group in gender and maybe healthcare type.  This would strengthen your conclusions.

Response: We appreciate the suggestions. We have examined non-responders and found some minors differences relative to responders with regards to age and education (gender, professional category etc was not different). We have added a discussion of this to the manuscript (lines 319-323).

Reviewer 3 Report

This is a well-designed and well-written manuscript to validate a transcultural version of WAI for Spain. The internal consistency, predictive validity, discriminant validity and constructive validity shown satisfactory result for this WAI Spanish version. But I have some concerns about the generalizability for various heath care works with different working demand, professional skill requirement and working type, such as physician, nurse, occupation therapist, pharmacist, etc., and even though to other professional field workers. The second, the most participants in the study were high educational level, university or more. Please discuss about the suitability of this WAI version apply to other different educational level, especially low educational level.

Author Response

We appreciate the comments and suggestions

Point 1:  I have some concerns about the generalizability for various health care works with different working demand, professional skill requirement and working type, such as physician, nurse, occupation therapist, pharmacist, etc., and even though to other professional field workers.

Response: We appreciate this comment. We acknowledge throughout the discussion and explicitly on lines 314-316 that generalisability of the findings is limited and outcomes can only be applied to the specific context of healthcare workers in Spain. It would be interesting for future work to address the aspects proposed by the reviewer and we have added this information to the manuscript on lines 316-317.

Point 2: The second, the most participants in the study were high educational level, university or more. Please discuss about the suitability of this WAI version apply to other different educational level, especially low educational level.

Response: We now discuss this issue on lines 319-322.

Round 2

Reviewer 1 Report

The work has improved substantially, although the conlcuioens are still quite poor.

Author Response

Response to Reviewer 1-round 2 Comments

Point 1: The work has improved substantially, although the conclusions are still quite poor

Response 1: We appreciate the suggestions. We have extended this section (see lines added lines 337-338 and 343-345).
